# Mental Health in Children, Adolescents, and Youths Living with Perinatally Acquired HIV: At the Crossroads of Psychosocial Determinants of Health

**DOI:** 10.3390/children10020405

**Published:** 2023-02-18

**Authors:** Álvaro Vázquez-Pérez, Carlos Velo, Luis Escosa, Teresa García-Lopez, Jose I. Bernardino, Eulalia Valencia, Rafael Mican, María José Mellado, Talía Sainz

**Affiliations:** 1Department of Pediatrics, Virgen de las Nieves University Hospital, 18014 Granada, Spain; 2Gregorio Marañón Health Research Institute (IISGM), 28007 Madrid, Spain; 3Department of Pediatrics, Infectious and Tropical Diseases, La Paz University Hospital and La Paz Research Institute (IdiPAZ), 28046 Madrid, Spain; 4Centro de Investigación Biomédica en Red en Enfermedades Infecciosas (CIBERINFEC), C. de Melchor Fernández Almagro, 3, 28029 Madrid, Spain; 5HIV Unit, Infectious Diseases and Internal Medicine Department, La Paz University Hospital and La Paz Research Institute (IdiPAZ), 28046 Madrid, Spain; 6Facultad de Medicina, Universidad Autónoma de Madrid, C. Arzobispo Morcillo, 4, 28029 Madrid, Spain

**Keywords:** HIV, vertical transmission, mental health, psychosocial factors, children, adolescents

## Abstract

Here, we aim to describe mental health (MH) in a cohort of children, adolescents, and young adults living with perinatally acquired HIV (PHIV) in Spain and explore the treatment gap for mental disorders. We also aim to analyze the potential association between MH issues to psychosocial risk factors (PSRFs) and identify management priorities. We conducted a descriptive transversal study that included all cases of PHIV under follow-up in a reference hospital in Madrid. The study included patients undergoing follow-up in the pediatric outpatient clinic and youths transferred from pediatric to adult care units after 1997. Epidemiological, clinical, immunovirological, and treatment-related data were collected, including PSRF and adverse childhood experiences (ACEs). Of the 72 patients undergoing follow-up, 43 (59.7%) had already been transferred to the adult outpatient clinic. The patients’ median age was 25 years (IQR 18–29), and 54.2% were women. Most patients were undergoing treatment (94.6%) and were virologically suppressed (84.7%). Although MH issues were present in 30 patients (41.7%), only 17 (56.7%) had been referred for evaluation to the Department of Mental Health, and only 9 (30%) had received a MH diagnosis. PSRFs were common (32% of participants had at least one PSRF) and were associated with MH issues and adherence issues (all *p* < 0.05). A multidisciplinary approach to address the psychological factors and social determinants of health is urgently needed, particularly during important life development stages, such as adolescence.

## 1. Introduction

With the successful implementation of mother-to-child transmission prevention programs, the incidence of vertical HIV infection has decreased dramatically in countries with access to antiretroviral therapy (ART), and the median age of perinatally HIV-infected (PHIV) patients is increasing [1]. Most PHIV patients born in the 1990s are now reaching adulthood and are transferring to adult units. These patients are at risk of morbidities associated with HIV infection and ART toxicity and experience the related social stigmatization and psychosocial impact of HIV [2]. Complex socioeconomic situations combined with frequent adverse childhood experiences (ACEs) are common among PHIV patients [3] and may negatively affect mental health and HIV-related outcomes [4,5]. Adolescents and young adults living with PHIV display higher rates of MH diagnoses when compared with the general population [6], with a significant impact on health-related quality of life (HRQoL) [2]. Despite the elevated prevalence of MH problems in PHIV patients, data from various cohorts suggest that MH problems are undertreated [7]. With the inclusion of HRQoL as the fourth 90 of the Joint United Nations Programme on HIV/AIDS (UNAIDS) 90-90-90 target, the treatment gap for mental disorders is now in the limelight [7,8]. To better understand the MH treatment gap among individuals living with PHIV, we describe MH, the prevalence of PSRFs, and the access to specific care in a cohort of patients with PHIV in Spain, so as to identify management priorities.

## 2. Materials and Methods

A cross-sectional study was conducted in a tertiary public hospital in Madrid, Spain. All patients with PHIV undergoing follow-up in December 2018 were included, irrespective of age, including patients managed by the pediatric outpatient clinic and those who had already transferred to adult care. All participants were included in the National Spanish Cohort of HIV-infected Children (CoRISpe) or the cohort of pediatric patients with HIV transferred to adult care (FARO project). Epidemiological, clinical, immunovirological, and treatment-related data were obtained from the CoRISpe–FARO project. CoRISpe has collected epidemiological, clinical, immunological, virological, analytical, and antiretroviral prospective data from HIV-infected children and adolescents, with follow-up in Spanish pediatric units, since 2008, as well as retrospective data since 1995. After the transition to adult units, these patients integrate the transition FARO cohort and continue to be followed prospectively [9].

The MH related variables included: MH referral and diagnosis, emotional disorders and/or behavioral problems, and management and treatment, including MH-related admissions. To obtain these variables of interest, we reviewed the participants´ medical records and then conducted semi-structured interviews with all health practitioners involved in the participants´ clinical care, including physicians, nurses, and psychologists. We collected all available data, whether identified by the patient, family, or health-care providers. Variables described in the literature as psychosocial risk factors (PSRFs), some of which are considered ACEs [3], were simultaneously collected and included: intrafamily conflict (parental separation or divorce), parental loss, institutionalization, foster care, and exposure to violence (domestic violence, bullying). Data on drug use and academic performance were also collected. Treatment adherence issues were collected retrospectively according to the medical records and based on managing clinicians. Viral suppression was defined as a viral load <50 copies/mL.

All participants signed informed consent/assent before inclusion in the CoRISpe-FARO cohorts. The study protocol was reviewed by the Ethics Committee of the participating centers (2008/0268, Hospital 12 de Octubre Ethics Committee, and amendment to include follow-up data after transitioning to adult clinics on 25 February 2014). We employed descriptive statistics to summarize the demographic and clinical characteristics of patients with and without MH issues. The qualitative data are reported in absolute frequencies and percentages, and the quantitative data are reported with medians with lower and upper quartiles (IQR). The chi-square test and Fisher’s exact test were employed to compare independent groups of categorical data. For quantitative data, normality was assessed using a Shapiro–Wilk test, and Mann-Whitney U tests were performed to address the association with MH disorders. The significance level for all analyses was set at 0.05. All analyses were performed using SPSS v.21.0 (SPSS Inc., Chicago, IL, USA).

## 3. Results

Of the 72 patients undergoing follow-up in December 2018, 43 (59.7%) had been transferred to the adult outpatient clinic. Overall, the median age was 25 years (IQR 18–29) and 54.2% were women. Most participants were born in Spain (83.3%) and had Spanish parents (70%). Overall, 94.6% of the patients were undergoing ART and 84.7% were virologically suppressed. Seventeen (23.6%) patients were in Center for Disease Control clinical stage C. Two patients had a clinical diagnosis of HIV encephalopathy. Table 1 summarizes the main characteristics of the study participants.

According to the medical records, 30 patients (41.7%) had presented with signs or symptoms suggesting MH issues, although only 17 of them (56.7%) had ultimately been referred for evaluation to the Department of Mental Health; 9 (30%) had a MH diagnosis and 9 (30%) were undergoing psychological therapy. The most common MH-related diagnoses among the patients evaluated by a MH specialist were anxiety (4/9), mood disorders (4/9), attention-deficit/hyperactivity disorder (2/9), borderline personality disorder (1/9), and anorexia nervosa (1/9). Two patients (2.8%) had been admitted due to MH disorders, including a suicide attempt and eating disorders.

MH problems were more prevalent among the 29 participants under pediatric follow-up, compared with the patients who had already transferred to adult clinics (62.1% vs. 27.9%; *p* < 0.05). We found no differences when comparing the management between the pediatric and adult clinics regarding the rate of referral to MH units or the establishment of a MH diagnosis. The prevalence of PSRFs in the cohort was 32%, but only three patients (4%) presented four or more PSRFs. Patients with MH issues more frequently had at least one PSRF (66.6% vs. 7.14%; *p* < 0.05). When restricting the analysis to the subgroup of patients with a MH diagnosis, six out of nine (66.6%) presented at least one PSRF. A total of 100% of patients under pediatric follow-up with a MH diagnosis (4/4) presented at least one PSRF.

Adherence issues were reported in 14 patients (19.4%), with most (91.7%) having at least one PSRF (*p* < 0.05). The prevalence of MH issues in the patients’ subgroup with adherence issues was very high (85.7%).

## 4. Discussion

In this study addressing MH in a cohort of patients living with PHIV, some of whom had already been transferred to adult units, the prevalence of MH problems was high (41.7%). PSRFs and ACEs were frequent in this population and were associated with MH problems. The referral to MH units was not common (23.6%), even among those presenting with signs and/or symptoms, and fewer than a third of the patients identified with MH issues had received a psychiatric diagnosis and were undergoing therapy. Our study highlights a significant treatment gap regarding MH for patients living with HIV. Addressing the social determinants of health and including MH and psychological support when managing this unique population from childhood to adulthood is essential for increasing patients’ quality of life.

Pediatric and adolescent MH is a global concern. According to a recently published report by the United Nations on pediatric MH, more than 13% of adolescents worldwide aged 10–19 years have a mental disorder [7]. In Western countries (and even more so after the COVID-19 pandemic), the number has risen exponentially [7]. Compared with the general population, those children, adolescents, and young adults who live with PHIV in various contexts show higher rates of MH issues [6]. Our data are in line with these findings, which suggest a prevalence of mental disorders up to 50% in certain studies [2,5,8,9,10,11]. The prevalence of PSRFs and ACEs was also high, with 32% of the patients having at least one and 4% having four or more. Although concerning, our numbers regarding ACE prevalence are smaller than the numbers documented in other studies, including in research on individuals living with HIV and the general population [3,4,5], with approximately 60% of the study participants experiencing at least one ACE and over 10% experiencing four or more ACEs. In our study, data regarding PSRFs were collected retrospectively; due to the retrospective nature of the study, specific questionnaires were not administered. The rate of PSRFs and ACEs might therefore be underestimated in this series. The fact that, during childhood, patients are accompanied by other family members during visits makes it easier to obtain a full picture of the social and family situation. Most pediatricians include data regarding the main caregivers in the medical records. However, studies have suggested that most children who experience ACEs take years to relate to them and others never do so. As indicated in previous studies, PSRFs were associated with adherence issues and MH problems [3].

Our definition for MH issues was broad, and based on retrospective records that were not always verified by a MH specialist, which is a significant limitation of our study. However, the fact that the patients were not systematically referred to MH units for evaluation, together with the low proportion of patients referred among those presenting with symptoms, suggests that overall, the prevalence of MH-related issues could be underdiagnosed and underestimated in our series. These data highlight the need for integrating psychological support for patients from a young age into health-care systems in order to to reduce the treatment gap regarding MH problems and improve the quality of life of individuals living with HIV. In fact, previous data suggest that adherence and behavioral problems during adolescence are a marker of poor outcomes during adulthood [12]. Worldwide, access to MH care is limited [7]. Although a significant proportion of adolescents across the globe live with mental health problems, service access is limited [7]. Among other barriers, there is a strong stigma associated with MH. Studies reveal that even in the developed nations, parents and adolescents often underuse potentially accessible mental health resources [7]. When analyzed, typical barriers include stigmatization, fear of embarrassment, lack of knowledge, and a lack of understanding of the help-seeking process [7].

Using a broad definition of MH, which includes all recorded signs and symptoms and not only the final diagnosis and/or treatment, our study found a higher rate of MH among patients undergoing pediatric follow-up. In our series, MH disorders were identified by the patient, family, and/or healthcare providers. The subjective perception of MH disorders were identified by the patient, family, and/or healthcare providers. The subjective perception of MH disorders could be influenced by adult–child relationships and power imbalances, including elements of labeling and stereotyping. Moreover, the transition from pediatric to adult care services in our series appeared to be significantly delayed. The median age of the patients in pediatric care at the time of their last visit was 16 years (IQR, 11–24), and 69% were 15 years of age or older. As indicated in previous studies addressing the transition [12], complicated patients and those with adherence issues tend to be followed-up for longer in pediatrics, in an attempt to increase patient autonomy, achieve full adherence, and obtain viral suppression before the transition. Patients with chronic conditions diagnosed during childhood face a challenging period upon reaching adolescence and during the transition from pediatric to adult care [13]. Several studies addressing various chronic conditions have shown how moving from a well-known, more personalized, youth-friendly outpatient model to a system that assumes patient autonomy might negatively affect treatment adherence and the link to care [13]. Most guidelines recommend an individualized plan for transitioning youth living with HIV to adult care [13,14] and the establishment of the optimal point in time according to clinical stability and patient autonomy and not to chronological age. Therefore, youths with PHIV are often transitioned at an older age [6,14]. This fact might account for the differences when comparing the prevalence of MH issues in pediatric and adult clinics.

Our study has several limitations. The retrospective design did not include a systematic psychological evaluation or the active collection of data regarding social determinants of health. The study had a small sample size and significant patient heterogeneity. An important limitation was the inclusion of a broad definition of MH, based on the clinical judgement of managing clinicians and not restricted to cases diagnosed after a proper MH assessment. However, mental diagnoses are complex and are often established late during follow-up, and a more restrictive definition would have undercut our ability to detect mild and moderate MH-related problems.

Larger, prospective studies are needed to identify the optimal strategies for managing this population. While unraveling the complex interplay between MH, HIV, ART, comorbidities, ACEs, and the social determinants of health, our results suggest that healthcare systems should include psychological evaluation and support in routine clinical practice when managing chronic conditions, such as HIV infection, to identify patients at risk since childhood and offer a multidisciplinary approach to address their medical and psychological issues, adherence and social determinants of health. To address the MH-related treatment gap is essential for increasing the quality of life of people living with HIV.

## Figures and Tables

**Table 1 children-10-00405-t001:** Characteristics of the study participants.

	Total N = 72 (100%)	MH Issues N = 30 (41.7%)	No MH Issues N = 42 (59.3%)	*p*
Age (years)				0.504
<12	7 (9.7)	3 (10.0)	4 (9.5)	
12–18	9 (12.5)	5 (16.6)	4 (9.5)	
>18	56 (77.8)	22 (73.3)	34 (80.9)	
Transferred to Adult Unit	43 (59.7)	12 (40.0)	31 (73.8)	0.004
Female	39 (54.2)	15 (50.0)	24 (57.1)	0.549
Born in Spain	60 (83.3)	24 (80.0)	36 (85.7)	0.521
Mother born outside Spain	16 (22.2)	8 (26.7)	8 (19.0)	0.397
Father born outside Spain	21 (29.2)	10 (30.0)	11 (26.1)	0.511
On ART				
Including PI	31 (44.9)	11 (36.7)	20 (47.6)	0.355
Including NNRTI	20 (28.9)	11 (36.7)	9 (21.4)	0.155
Including INSTI	31 (44.9)	12 (40.0)	19 (45.2)	0.658
HIV VL < 50 copies/mL	61 (84.7)	26 (86.6)	35 (83.3)	0.753
Median CD4 counts (cells/μL) *	823	905	780	0.068
	(596–1166)	(716–1252)	(548–1036)	
CD4 Nadir (cells/μL) *	285	319	210	0.149
	(108–456)	(96–564)	(102–415)	
CDC Stage C	17 (23.6)	8 (26.6)	9 (21.4)	0.646
Psychosocial Risk Factors				
Parental loss	8 (11.1)	8 (26.6)	0 (0)	<0.001
Adoption	8 (11.1)	6 (20.0)	2 (4.7)	0.06
Intrafamily conflict	17 (23.6)	16 (53.3)	1 (2.3)	<0.001
Institutionalization	3 (4.2)	3 (10.0)	0 (0)	0.068
Alcohol	3 (4.2)	3 (10.0)	0 (0)	0.068
Cannabis	3 (4.2)	2 (6.6)	1 (33.3)	0.567
Bullying	3 (4.2)	3 (10.0)	0 (0)	0.068
Low academic performance	13 (18.0)	13 (43.3)	0 (0)	<0.001
1 or more PSRFs	23 (32)	20 (66.7)	3 (7.1)	<0.001
4 or more PSRFs	3 (4.2)	3 (10.0)	0 (0)	0.068
Adherence problems	14 (19.4)	12 (40.0)	2 (4.7)	0.004

* Median [IQR]. ART, antiretroviral therapy; CDC, Centers for Disease Control; INSTIs, integrase inhibitors; MH, mental health; MH Issues: patients with MH alterations (mental health diagnosis, emotional disorders, and/or behavioral problems), whether identified by the patient, caregivers, or health-care providers; NRTIs, nucleoside reverse transcriptase inhibitors; NNRTIs, non-nucleoside reverse transcriptase inhibitors; PI, protease inhibitors; PSRF, psychosocial risk factor; VL, viral load.

## Data Availability

Data are available on request due to privacy/ethical restrictions. Data are available on request from the authors.

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
