# Peer review of "Mental Health in Children, Adolescents, and Youths Living with Perinatally Acquired HIV: At the Crossroads of Psychosocial Determinants of Health"

_children, 2023, doi:10.3390/children10020405_

Round 1

Reviewer 1 Report

Thank you very much for an interesting article submitted for review. From the point of view of a person working with HIV-infected children, such an assessment is extremely interesting.

The aim of the study is to identify risk factors for MH diseases. These factors are not presented in the text. Unfortunately, in the opinion of the reviewer, the goal of the work was not achieved. Risk factors have not been detailed. The conclusions apply to the adult and adolescent population.

Work needs improvement. In the opinion of the reviewer, the selection of the group is clear, but the recorded disturbances have poor credibility. It is worth improving this part. Merely relying on observations regarding mental disorders contained in the documentation is not very credible.

Line 86: The inclusion criteria for the analysis are unclear. I propose to improve.

line 92: It is not clear why the authors classify 30 patients as MH, even though only 17 were examined in a specialist center.

line 94: The description of the disorders is very general. There is no information on who diagnosed them.The reviewer is not convinced that diagnosing MH disorders only on the basis of observation conducted by the environment is sufficiently reliable. It would be much more interesting to explain what was found in reference centers. The diagnosis of the other 13 people is unreliable.

line 99: The study group is heterogeneous. Comparing MH disorders in young children (under 12 years of age) and adults is not correct.

line 107 "all the youths"  Youths is the plural form of youth in the sense of a singular boy or young man. Do you mean that?

line 110 Table1. Please explain at what age patients are referred to an adult center. The table shows that people over 18 were still under the care of the children's centre. In my opinion, the exact explanation of when patients cross over to adult centers requires clarification. It is the basis of later statistics showing a higher incidence of MH disorders in the pediatric population.

line 117 Is the prevalence rate of disorders in society known? It's worth refilling.

line 122 Wouldn't it be worth taking care of children, since in this group the rate was the highest?

line 150 This fragment does not match the table. I think that the lack of specifying the moment of transfer to the center for adults results in ambiguities in the further parts of the text. Did the authors take into account that in the case of children, the interview is most often collected from the family, which affects its credibility. Comparing it with an interview collected from adults is, in the opinion of the reviewer, impossible. We collect the interview directly from the adult. There is a greater possibility of withholding certain facts.

In the opinion of the reviewer, the work needs improvement primarily in terms of the description of the research methods used and the selection of the group.

Author Response

Thank you very much for an interesting article submitted for review. From the point of view of a person working with HIV-infected children, such an assessment is extremely interesting.

The aim of the study is to identify risk factors for MH diseases. These factors are not presented in the text. Unfortunately, in the opinion of the reviewer, the goal of the work was not achieved. Risk factors have not been detailed. The conclusions apply to the adult and adolescent population.

We agree with the reviewer that our study has several limitations, starting with its retrospective design.  We aimed to describe how Mental Health (MH) disorders were diagnosed and managed in a cohort of children, adolescents and young adults living with perinatally acquired HIV (PHIV) in Spain, underlying the relevance of addressing psychosocial risk factors (PSRF). The main aim has been reformulated and we have clarified the methodology and definitions to avoid misinterpretations. We understand that the conclusions apply to all perinatally HIV-infected patients. Many of the individuals included in our study were already young adults, but we included the information relative to childhood. 

Work needs improvement. In the opinion of the reviewer, the selection of the group is clear, but the recorded disturbances have poor credibility. It is worth improving this part. Merely relying on observations regarding mental disorders contained in the documentation is not very credible.

We agree that it would have been optimal to evaluate all participants using a standardized methodology. Unfortunately, this is not the case in a retrospective descriptive study, and we have to stick to the information registered in medical records. We present disaggregated data on referrals to mental Health and Neurology, the drugs prescribed if any, and psychotherapy…but we discuss the probable underdiagnosis due to a lack of mental health assessment in routine clinical practice. This is in fact one of the main conclusions of our work: the treatment gap in mental health. We appreciate the comment as we have reformulated the aims and discussion to focus on that. The study design limits our possibilities, and this is well-recognized among the limitations in the discussion section.

Line 86: The inclusion criteria for the analysis are unclear. I propose to improve.

The methods section has been revised to clarify the inclusion criteria.  

Line 92: It is not clear why the authors classify 30 patients as MH, even though only 17 were examined in a specialist center.

This is an interesting point, as one of the main points of interest of our work (and the main limitation as well) is the lack of standardized evaluations performed in this cohort in a European context. Mental Health is not included in routine clinical practice, even when signs and symptoms are registered in medical records. In Spain, there is still stigma related to mental health services, and this can be one of the underlying reasons why a referral to MH units was infrequent (23.6%). Once referred and evaluated, diagnoses are often complex and many patients remain under follow despite not having a proper diagnosis. In our series, less than one-third of the patients evaluated and followed and/or treated at MH facilities had a psychiatric diagnosis. We decided to include all patients referred, evaluated, treated or under follow-up in our definition of mental health issues, although we present disaggregated data on all the variables considered. 

Line 94: The description of the disorders is very general. There is no information on who diagnosed them. The reviewer is not convinced that diagnosing MH disorders only on the basis of observation conducted by the environment is sufficiently reliable. It would be much more interesting to explain what was found in reference centers. The diagnosis of the other 13 people is unreliable.

We are convinced that both definitions are important, although we understand the concerns regarding the reliability of a more inclusive definition of mental health issues. Definitions have been revised and clarified in the Methods section to avoid misunderstandings. A broad definition including all mental health diagnoses, emotional disorders, and/or behavioral problems identified by health care provided and registered in medical records but not necessarily evaluated by mental health professionals has been used in the table, as we understand that using a restrictive definition will lead to a significant underestimation. However, we recognize these limitations in the discussion section. Information regarding referral to MH units, treatments, or admissions due to MH-related diagnoses has also been collected.

Line 99: The study group is heterogeneous. Comparing MH disorders in young children (under 12 years of age) and adults is not correct.

We understand the reviewers’ concern. The study cohort includes all PHIV under follow-up in our center. Adolescence is a key period in terms of health, and many comorbidities associated to HIV are diagnosed in adulthood. We intended describe the whole spectrum of diseases, and therefore included also young adults. We did not intend to compare children vs adults, but to compare the management of mental health in pediatrics and adult clinics. Some sections have been reformulated to clarify this point.

Line 107 "all the youths" Youths is the plural form of youth in the sense of a singular boy or young man. Do you mean that?

The erratum has been corrected. Thank you

Line 110 Table1. Please explain at what age patients are referred to an adult center. The

table shows that people over 18 were still under the care of the children's centre. In my opinion, the exact explanation of when patients cross over to adult centers requires clarification. It is the basis of later statistics showing a higher incidence of MH disorders in the pediatric population.

The reviewer has raised a very interesting point. According to most guidelines, the transition is a process that needs to be adjusted individually. Our protocol includes a checklist to ensure that different aspects (patients’ autonomy, linkage to care, immunizations update, comorbidities screening…) are addressed during the transition, but does not establish a threshold for referral to adults. Therefore, complex patients often remain in pediatrics until an older age. We discuss the effects of these particularities in the discussion.

Line 117 Is the prevalence rate of disorders in society known? It's worth refilling:

We sincerely appreciate the comment. The suggested information and a new paragraph discussing prevalence have been included. This point is also discussed when explaining that our data are in line with previous publications addressing the population of PHIV in different contexts, suggesting a prevalence of mental disorders higher than the one found in pour series (up to 50%).

line 122 Wouldn't it be worth taking care of children, since in this group the rate was the highest?

We sincerely appreciate the comment as this is a key point. We have emphasized this idea.

line 150 This fragment does not match the table. I think that the lack of specifying the moment of transfer to the center for adults results in ambiguities in the further parts of the text.

We have clarified the aspects related to the transition from pediatric to adult care, which in our series happens rather late. The manuscript now includes references to previous studies addressing transition that have also underlined this fact.

Did the authors take into account that in the case of children, the interview is most often collected from the family, which affects its credibility. Comparing it with an interview collected from adults is, in the opinion of the reviewer, impossible. We collect the interview directly from the adult. There is a greater possibility of withholding certain facts.

We totally agree with the reviewer here, this is clearly described in the Material and Methods and discussed as a limitation in the discussion. However, to our understanding the main risk is the probable bias in terms of a higher perception of mental health problems among the families in comparison to the patients.  Once again, this is a limitation inherent to the study design, which we can discuss but cannot change in a retrospective study.

In the opinion of the reviewer, the work needs improvement primarily in terms of the description of the research methods used and the selection of the group.

We believe that the manuscript has improved with all reviewers’ comments and sincerely hope you find it suitable for publication now. Unfortunately, the limitations inherent to the study design can be discussed but not modified.

Reviewer 2 Report

An interesting clinical paper.

The legend on table is short. They need to be more descriptive and to explain all shortcuts, as physicians and scientists easy understand the paper. And if the descriptions in the legends are not sufficient, they may not understand the paper.

Author Response

An interesting clinical paper.

We thank you for the recognition of mental health as an important problem.

The legend on table is short. They need to be more descriptive and to explain all shortcuts, as physicians and scientists easy understand the paper. And if the descriptions in the legends are not sufficient, they may not understand the paper.

We appreciate the comment and have improved the legend explaining all shortcuts.

Reviewer 3 Report

This manuscript by Pérez A. V et.al investigates mental health (MH) disorders in a cohort of children, adolescents, and young adults living with perinatally acquired HIV (PHIV) in Spain to analyze their potential relation to psychosocial risk factors (PSRF), and to identify management priorities

Few minor concern

1.     Ethical numbers are missing.

2.     In the table, please revisit the way the percentage is represented. It should be (.) instead of (,).

3.     Extensive editing of the English language and style is required

Author Response

This manuscript by Pérez A. V et.al investigates mental health (MH) disorders in a cohort of children, adolescents, and young adults living with perinatally acquired HIV (PHIV) in Spain to analyze their potential relation to psychosocial risk factors (PSRF), and to identify management priorities

Few minor concern

  1. Ethical numbers are missing.

We appreciate the comment, Ethical approval number has been included.

  1. In the table, please revisit the way the percentage is represented. It should be (.) instead of (,).

Thank you, it has been revised and unified.

  1. Extensive editing of the English language and style is required

The manuscript has been revised by the institute's Scientific English Language specialists

Round 2

Reviewer 1 Report

I think the authors made the great work to improve the article.